

# Results on high energy galactic cosmic rays from the DAMPE space mission

Leandro Silveri[1,2]⋆, on behalf of the DAMPE Collaboration

1 Gran Sasso Science Institute (GSSI), Via Iacobucci 2, I-67100 L'Aquila, Italy
2 Istituto Nazionale di Fisica Nucleare (INFN) - Laboratori Nazionali del Gran Sasso, Italy

⋆ leandro.silveri@gssi.it

## Abstract

**DAMPE (Dark Matter Particle Explorer) is a satellite-born experiment launched in 2015 in a sun-synchronous orbit at 500 km altitude, and it has been taking data in stable conditions ever since. Its main goals include the spectral measurements up to very high energies, cosmic electrons/positrons and gamma rays up to tens of TeV, and protons and nuclei up to hundreds of TeV. The detector's main features include the 32 radiation lengths deep calorimeter and large geometric acceptance, making DAMPE one of the most powerful space instruments in operation, covering with high statistics and small systematics the high energy frontier up to several hundreds TeV. The results of spectral measurements of different species are shown and discussed.**



## 1 Introduction

Satellite-borne missions in the last decade have achieved unprecedented improvements both in precision and energy sensitivity for direct measurements of cosmic rays. This has unveiled unexpected results, such as the observation of a break from single power-law behaviour even prior to the knee region in multiple nuclei spectra. Among the experiments currently in orbit, DAMPE [1] is the one capable of reaching the hundreds of TeV energy with good statistics because of its high acceptance and the deep calorimeter. This makes it possible to compare the results with some ground-based CR detection facilities. The acceptance for electrons is $\approx 0.3$ m$^2$ sr for energies above $\approx 10$ GeV.

## 1.1 DAMPE space mission

DAMPE was launched on December 17, 2015, from the Jiuquan Satellite Launch Center, China, in a sun-synchronous orbit at 500 km altitude. The mission goals are:

- Very accurate measurements of cosmic-ray spectra of electrons, protons and nuclei;

- Perform $\gamma$-ray astronomy;

- Measure $\gamma$ spectral lines that could point to possible Dark Matter self-interaction channels.

## 1.2 DAMPE sub-detector modules

The DAMPE detector is made out of 4 sub-detectors (fig. 1), each one having a different task in order to successfully discriminate the nature of the impinging particles, track their direction and measure their energy.

More specifically, starting from the top and going toward the bottom of the detector like shown in fig. 1:

- Plastic Scintillator Detector: it is made by 4 layers of plastic scintillator bars used to perform charge measurement and gamma anticoincidence, two on each view (X and Y) in order to reconstruct the information of the impact point, and staggered to improve hermeticity;

- Silicon-Tungsten Tracker: it consists of 6 planes of silicon microstrip detectors, each one featuring an X and Y segmented layer, with some of them interleaved with tungsten plates in order to enhance the gamma conversion probability for gamma tracking. This results in an angular resolution at normal incidence at 100 GeV of $\approx 0.1^{o}$;

- BGO Imaging Calorimeter: it embodies 14 layers of BGO bars, and the bars are positioned inside a single layer along the X or the Y direction of the detector reference frame, alternating them in such a way that allows for the possibility of reconstructing the image of the shower, as well as performing the energy measurement. The obtained shower image makes it possible to distinguish hadronic from electromagnetic showers. The total depth of this calorimeter is 32 $X_0$ (radiation lengths) and 1.6 $\Lambda_I$ (interaction lengths) ;

- Neutron Detector: it is composed of 4 tiles of boron-loaded plastic scintillator placed in a single layer, allowing for the possibility to detect neutrons, providing a further discrimination power against electromagnetic showers.

A more detailed description is given in [1].

# 2 Proton and Helium results

Using the data collected by DAMPE from January 2016 to June 2018 (30 months), it was possible to perform the measurement of the proton spectrum from 40 GeV to 100 TeV [2]. The result confirmed a hardening which was already observed by other experiments at 500 GeV [3] [4] [5] [6] [7]. Measurements also show a softening at 14 TeV with a strong evidence of 4.7 $\sigma$ significance. These breaks pointed out that a broken power law model describes data in a more appropriate way when compared to the single power law spectrum.

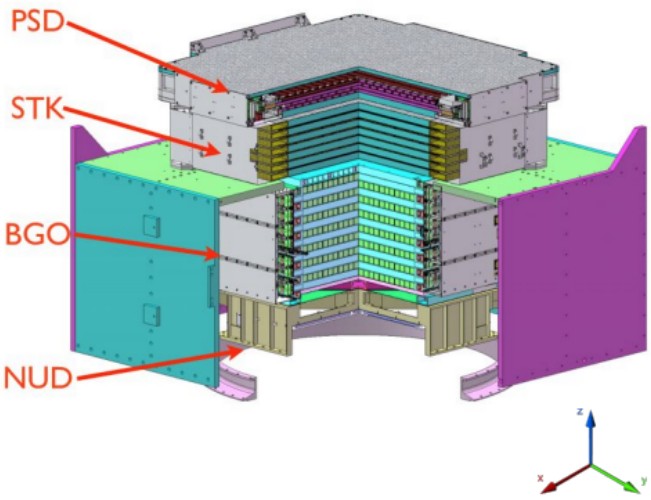

Figure 1: DAMPE with its sub-detector modules, from top to bottom: PSD (Plastic Scintillator Detector), STK (Silicon-Tungsten tracKer), BGO (BGO Imaging Calorimeter) and NUD (NeUtron Detector).

A similar result was also obtained for helium: the DAMPE He spectrum was obtained using 54 months of data (from January 2016 to June 2020), and it has been published for the energy range starting from 70 GeV up to 80 TeV of total kinetic energy [8]. It confirmed a hardening which was previously observed by other experiments [4] [5] [6] [9] [10] and showed a very strong evidence for a softening at 34 TeV with 4.3 $\sigma$ significance, making a broken power law model the preferred model in this case as well. This result, when combined to the proton spectrum, suggests that the spectral features are more likely to be rigidity-depending rather than energy-depending, even though the latter cannot be ruled out because of the uncertainties.

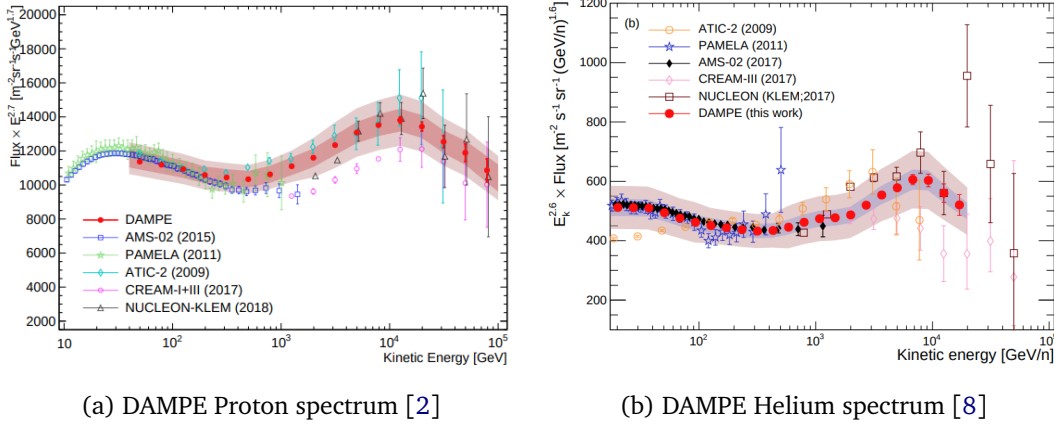

(a) DAMPE Proton spectrum [2]

(b) DAMPE Helium spectrum [8]

Figure 2: DAMPE proton (a) and Helium (b) spectra, compared with measurements from various experiments ( [3] [4] [5] [6] [7] and [4] [5] [6] [9] [10] respectively).

In order to have more details about the behaviour of cosmic-ray spectrum at the highest direct measured energies, the combined analysis of protons and helium is being carried out [11]. This analysis is characterised by a very high statistical sample with very low contamination, since the flux of the neighbouring nuclei is significantly lower, resulting in a high-energy spectrum that can also be compared with indirect measurements, both for the type of measurement

and the energies. In particular, the preliminary results shown in fig. 3 are already comparable to the ones provided by HAWC [12], and getting close to the energies of the first part ARGO-YBJ [13] flux. In the future, we might be able to reach an energy high enough to have our measurement almost comparable to KASCADE flux [14] as well. Aside from the importance of the measurement per se, this spectrum might be very useful as a bridge between direct and indirect measurements, since the former are not depending on uncertainties related to Extensive Air Shower: in particular the underlying hadronic interaction models.

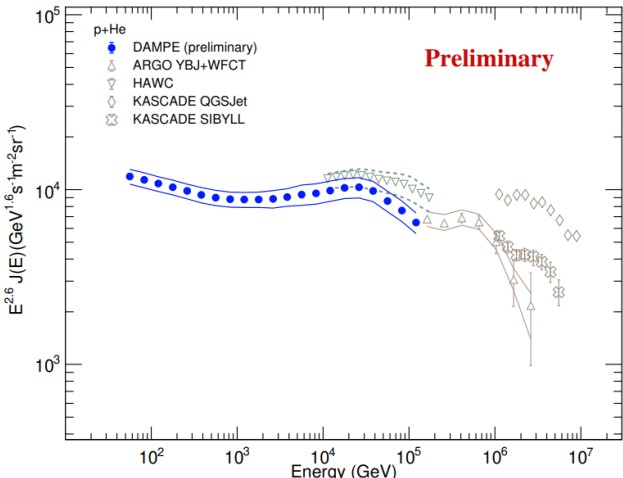

Figure 3: DAMPE preliminary results of p+He spectrum [11], compared with HAWC [12] and more indirect measurements [13] [14]. The DAMPE energy range is approaching indirect experiments.

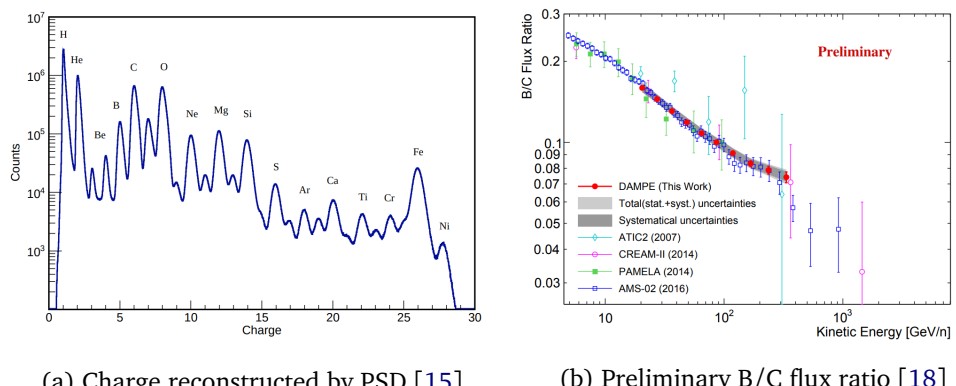

(a) Charge reconstructed by PSD [15]

(b) Preliminary B/C flux ratio [18]

Figure 4: The DAMPE charge resolution given by the PSD (a) is precise enough to distinguish most of the peaks related to the CR composition up to Nickel. As an example, the preliminary Boron over Carbon flux ratio is shown (b).

# 3 More analyses on heavier nuclei

Several additional works on heavier nuclei are in progress with the aim of measuring their flux and the ratio among them. The nuclei are identified by the energy deposit inside the PSD [15], with an example histogram of reconstructed charge from PSD shown in fig. 4a. Currently, the

ongoing analyses include:

- Single species spectra of light and intermediate mass nuclei, such as Boron and Carbon [16];

- Spectra of heavy elements, like Fe [17] and ultra-Fe species;

- Flux ratios, like the Boron over Carbon [18] shown in fig. 4b.

## 4 Conclusion

The DAMPE mission has been continuously collecting data since December 2015, in stable conditions and with all its sub-detectors fully working. Data analysis produced already important results, like the proton [2] and helium [8] spectra shown in this work, which highlighted new features in the cosmic-ray spectra, and the electrons [19] and $\gamma$ [20] as well. Further analyses on other nuclei are currently being carried out.

## Acknowledgements

**Funding information** The DAMPE mission was funded by Chinese Academy of Sciences (CAS). Activities are supported by National Key Research and Development Program of China (2016YFA0400200), National Natural Science Foundation of China (11921003, 11622327, 11722328, 11851305, U1738205, U1738206, U1738207, U1738208, U1738127), strategic priority science and technology projects of CAS (XDA15051100), 100 Talents Program of CAS, Young Elite Scientists Sponsorship Program by CAST (YESS20160196), Program for Innovative Talents and Entrepreneur in Jiangsu, the Swiss National Science Foundation (SNSF), National Institute for Nuclear Physics (INFN), Italy, and European Research Council (ERC) Horizon 2020 programme (851103).

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
