# Peer review of "Results on high energy galactic cosmic rays from the DAMPE space mission"

_SciPost Physics Proceedings, doi:SciPost Phys. Proc. 13, 012 (2023)_

## Round 1 · Referee Report · Anonymous (Referee 1) · 2022-9-3

Strengths

  • good, valid and interesting results

Weaknesses

  • could be a little more concrete and better linguistically

Report

The paper shows interesting results exactly fitting the target of the ISVHECRI conference. The content is valid for publication and I congratulate the entire DAMPE collaboration to this results and the sucessful experiment.
What follows are a list of some proposed corrections, questions and comments where I ask the author to treat before publication.

Requested changes

Abstract: - "..up to very high energies": please specify in the abstract what you mean with very high energies. - last sentence: would better read as: The results of spectral measurements of different species are shown and discussed.

section 1: - first sentence "....for direct measurements of cosmic rays." - what do you mean with "mulitple spectra"? of different experiments, of different masses, of ...? - again "highest energy": not clear how you define highest energy

section 1.1: - ...of cosmic-ray spectra of electrons, protons and nuclei; - ... \gamma-ray - ... self-interaction channels

section 1.2: - either sub-detector or subdetector, please do not mix it - why radiation lengths and interaction lengths is written in italic? (not needed) - please specifiy what you mean with "since neutrons cannot be produced in there." where thesy cannot produced?

section 2: - Proton and Helium results (not "protons") - "simple power law" ==> "single power law" - do not know what you mean with "rather than energy-depending"? do you mean mass dependence compared to charge dependence? - "... the behaviour of cosmic rays at the highest...." - The sentence "This type of measurement..." reads a bit strange, may be it is better like "This analysis is characterised by a very high statistical sample with very low contamination, resulting in a high-energy spectrum, which can also be compared with indirect measurements." - can you specify a bit more what do you mean, what is the definition of "very low contamination"? - EAS models: in particular the underlying hadronic interaction models. - caption fig3: ...The DAMPE energy range is approaching....

section 3: - Presently ==> Currently - Spectra of heavy elements, like...

section 4: - can you put references behind ...teh electrons [] and \gamma [] as well.

References: - nice list of references , where you always have reviewed journal articles, except for [14]. Can you change this?

---

## Round 2 · Referee Report · Anonymous (Referee 1) · 2022-9-23

Report

The revised version is fulfilling all requests and is recommended to be published as it is.

---

## Editorial Decision

published